# Discovery of Cell Number-Interstitial Fluid Volume (CIF) Ratio Reveals Secretory Autophagy Pathway to Supply eHsp90α for Wound Healing

**DOI:** 10.3390/cells13151280

**Published:** 2024-07-30

**Authors:** Cheng Chang, Xin Tang, Axel H. Schönthal, Mei Chen, David T. Woodley, Yanzhuang Wang, Chengyu Liang, Wei Li

**Affiliations:** 1Department of Dermatology and the USC-Norris Comprehensive Cancer Centre, Los Angeles, CA 90089, USA; chan296@usc.edu (C.C.); tangxin@usc.edu (X.T.); chenm@usc.edu (M.C.); dwoodley@usc.edu (D.T.W.); 2Department of Molecular Microbiology & Immunology, University of Southern California Keck Medical Centre, Los Angeles, CA 90033, USA; schontha@usc.edu; 3Department of Molecular, Cellular and Developmental Biology, University of Michigan, 1105 North University Avenue, Ann Arbor, MI 48109, USA; yzwang@umich.edu; 4Molecular and Cellular Oncogenesis Program, The Wistar Institute, 3601 Spruce Street, Philadelphia, PA 19104, USA; cliang@wistar.org

**Keywords:** conditioned medium, interstitial fluid, tissue microenvironment, biomarkers, Hsp90, wound healing

## Abstract

Cell secretion repairs tissue damage and restores homeostasis throughout adult life. The extracellular heat shock protein-90alpha (eHsp90α) has been reported as an exosome cargo and a potential driver of wound healing. However, neither the mechanism of secretion nor the genetic evidence for eHsp90α in wound healing has been substantiated. Herein, we show that tissue injury causes massive deposition of eHsp90α in tissues and secretion of eHsp90α by cells. Sequential centrifugations of conditioned medium from relevant cell lines revealed the relative distributions of eHsp90α in microvesicle, exosome and trypsin-sensitive supernatant fractions to be approximately <2%, <4% and >95%, respectively. Establishing the cell-number-to-interstitial-fluid-volume (CIF) ratio for the microenvironment of human tissues as 1 × 10^9^ cells: 1 mL interstitial fluid enabled us to predict the corresponding tissue concentrations of eHsp90α in these fractions as 3.74 μg/mL, 5.61 μg/mL and 178 μg/mL. Remarkably, the 178 μg/mL eHsp90α matches the previously reported 100–300 μg/mL of recombinant eHsp90α whose topical application promotes maximum wound healing in animal models. More importantly, we demonstrate that two parallel secretory autophagy-regulating gene families, the autophagy-regulating (AR) genes and the Golgi reassembly-stacking protein (GRASP) genes work together to mediate the secretion of the physiological concentration of eHsp90α to promote wound healing. Thus, utilization of the CIF ratio-based extrapolation method may enable investigators to rapidly predict biomarker targets from cell-conditioned-medium data.

## 1. Introduction

Tissue injury and repair occur all the time and everywhere in the human body throughout life. During the healing process, cells secrete molecules around the injured site in response to the stress signals of an ischemic gradient that ranges from efficient nutrients and oxygenation to gradually near anoxia [1]. For instance, the microenvironment of a wounded area of skin quickly becomes hypoxic, with a shortage of nutrient supply, due to surrounding vascular disruption and high oxygen consumption by cells at the wound edge and in the granulation tissue [2,3,4]. Similar to wound healing, when a tumor expands too fast by outgrowing the nearby neovascularization by over 150 μm, cells inside the tumor encounter insufficient oxygen and nutrient supply. Under both pathological conditions, cells either turn on an intrinsic stress-responding mechanism to repair the damaged tissue or move away from the hazard environment to re-connect with the nearest blood vessel. If both attempts fail, cells turn on apoptotic programs and die [5].

The mechanisms of protein secretion for eukaryotic cells are largely divided into the classical secretion pathway, involving endoplasmic reticulum (ER) and Golgi apparatuses, and unconventional protein secretion (UPS) pathways. The ER/Golgi protein trafficking pathway requires the to-be-secreted proteins to present a N-terminal signal peptide, a 16–30-amino-acid-long peptide with a positively charged n-region, a hydrophobic h-region and a neutral c-region at the N-terminal of the proteins. The signal peptides are recognized by the signal recognition particle (SRP) on the ER surface and transported into the ER lumen. In ER, the signal peptides are cleaved off and proteins are stabilized by molecular chaperones and packed into COPII-coated vesicles and delivered into cis-Golgi. Next, proteins are transported through secretory vesicles (SV) to the plasma membrane and finally secreted to extracellular space [6]. However, a substantial number of secreted proteins lack a classical signal peptide and are called leaderless cargos. These proteins are secreted instead by one of the four UPS pathways including: (1) type I, plasma-membrane-pore-formation-mediated protein secretion; (2) type II, ABC-transporter-based protein secretion; (3) type III, organelle-based protein translocation and (4) type IV, Golgi-bypass [7,8,9]. In comparison to the classical protein trafficking pathway, the UPS pathways are less characterized and pose many unanswered questions.

Paracrine or cell-to-cell signaling via secreted molecules represents an evolutionarily conserved communication mechanism for tissue repair and homeostasis restoration [10]. The term ‘conditioned medium’, which is the source of the secreted molecules for laboratory research, was initially defined as the liquid phase of a cell-culture environment enriched with the secretome of cultivated tissue [11]. Today, cell-conditioned medium (CM) has become an irreplaceable source for research into new biomarkers and drug targets of human disorders such as systemic chronic inflammation, non-healing wounds and cancer [12]. However, the technology currently used to study secreted molecules in CM involves tedious, time-consuming and low-success processes, including protein concentration, mass spectrometry, ELISA verification, protein purification, RNAi library screening and individual gene characterization, just to mention a few, prior to reaching a final positive or negative answer [13]. For example, two independent laboratories were searching for a wound healing- and tumor invasion-promoting factor in CM of normal and cancerous cell lines using the conventional approach. Although ultimately successful, it took several years for the laboratories to identify the secreted form of heat shock protein-90alpha (Hsp90α) as a promotor of both wound healing and tumorigenesis [14,15]. The results of several studies further raised the level of interest in secreted Hsp90α by suggesting that exosomes and exomeres mediate secretion of Hsp90α in vitro [16,17,18]. However, few have thought about establishing a mechanism that bypasses the tedious and costly conventional approach and allows a rapid prediction of the physiological relevance of a target protein in CM.

In the current study, we utilized secreted Hsp90α or extracellular Hsp90α (eHsp90α), which have previously been demonstrated to promote wound healing and tumorigenesis at 100–300 μg/mL range in both murine and porcine wound models [19,20,21,22], as the protein markers to establish (1) a new mechanism that rapidly predicts the in vivo importance of a secreted protein in CM in vitro and (2) the secretory pathway that supplies the amount of eHsp90 for wound healing. We utilized the parameters of solid and liquid portions in the human body to calculate the cell-number-to-interstitial-fluid-volume (CIF) ratio for the in vivo tissue microenvironment. We demonstrate that application of the human CIF ratio to secreted Hsp90α from CM predicts in vivo concentration of secreted Hsp90α that perfectly matches the reported 100–300 μg/mL dosages for promoting maximum wound healing in vivo. More intriguingly, the CIF ratio mechanism allowed us to find that the type III unconventional protein secretion (UPS) pathway, called secretory autophagy, is responsible for supplying over 95% of secreted Hsp90α for wound healing. In contrast, the exosome-associated Hsp90α makes up no more than 20 μg/mL, insufficient for promoting wound healing in vivo. Thus, the discovery of the CIF ratio-based extrapolation method may enable investigators to dramatically shorten the screening and identification processes and to focus on a limited number of biomarker and drug targets.

## 2. Materials and Methods

### 2.1. Cell Lines and Cell Culture

The 293T and HeLa and human neonatal dermal fibroblast cells were cultured in regular Dulbecco’s modified Eagle medium (DMEM) and MDA-MB-231 in high glucose DMEM with penicillin–streptomycin (100 U/mL–0.1 mg/mL) and 10% fetal bovine serum (FBS). The third or fourth passage of the primary human neonatal dermal fibroblasts, was used in experiments. Immortalized human keratinocyte (IKC) were cultured in Epilife medium with added growth factor supplements (Thermo Scientific, Waltham, MA, USA). Hela GRASP55- and GRASP-65-knockout cell lines were provided by the laboratory of Dr. Yanzhuang Wang (University of Michigan, Ann Arbor, MI, USA). All cells were tested bi-monthly to ensure that they were mycoplasma-free by the paid service at USC Tissue Culture Core.

### 2.2. Reagents and Antibodies

TGF-α was purchased from Fitzgerald Industries International (Gardner, MA, USA). Trypsin (2.5%) was purchased from Thermo Fisher Scientific (Waltham, MA, USA). ECL Western blotting detection reagent (Cat. # RPN2106) was purchased from Amersham, Inc. (Marlborough, MA, USA). The antibodies used in this study include anti-Hsp90α antibody (NB120-2928), anti-CD9 (D3H4P;13403), anti-Atg5 (D5F5U; no. 12994T), anti-Atg7 (D12B11; no. 8558T), anti-Atg16L1 (D6D5; no. 3495T), anti-Beclin-1 (D40C5; no. 3495T) and anti-LC3A (D50G8; no. 4599T) antibodies from Cell Signaling Technology (Beverly, MA, USA); anti-β-actin antibody (AC038) from Transduction Laboratories (San Jose, CA, USA); and anti-GRASP55 (10598-1-AP) and anti-GRASP65 (66651-1-lg) from the Proteintech Group (Rosemont, IL, USA).

### 2.3. Two Lentiviral Vector Systems for Up- or Downregulation of Target Genes

The concentrations of packaging, overexpression and downregulation vectors used in this study were measured by Eppendorf Biospectrometer (Hamburg, Germany). The protocol for using lentiviral systems for up- or downregulation of gene of interest, including virus packaging, isolation, infection and analysis of gene expression were as previously described [23,24]. The lentiviral infection system pHAGEII-pEF2a was used to overexpress exogenous GRASP55 cDNA. The pHR-CMV-puro RNAi delivery system was used to deliver shRNA against genes of interest. The shRNA sequence for human GRASP55 was CCACCTGAAGACTTGTGTTAA (sense). shRNAs against human Atg5, Atg7, Atg16L1, Beclin-1 and LC3A in pGIPZ-puro RNAi delivery system were constructed in Dr. C-Y Liang’s laboratory, including CTTGGAACATCACAGTACA (sense) against Atg5, TGCTGTTGACAGTGAGCGACCAGCTATTGGAACACTGTATTAGTGAAGCCACAGATGTAATACAGTGTTCCAATAGCTGGGTGCCTACTGCCTCGGA (sense) against Atg7, TGCTGTTGACAGTGAGCGACCAACAGAACTTGATTGTAAATAGTGAAGCCACAGATGTATTTACAATCAAGTTCTGTTGGGTGCCTACTGCCTCGGA (sense) against Atg16L1, TGCTGTTGACAGTGAGCGAGCCAATAAGATGGGTCTGAAATAGTGAAGCCACAGATGTATTTCAGACCCATCTTATTGGCCTGCCTACTGCCTCGGA (sense) against Beclin-1 and GTCATTGTCCCTCTGCAGA (sense) against LC3A.

### 2.4. Fractionation of CM by Centrifugations

Cells were seeded at a concentration of approximately 2 million per 15 cm in a cell culture dish at 37 °C in a humidified incubator with 5% CO_2_ in DMEM culture medium with high glucose supplemented with 10% FBS and 1% P/S. When cell growth reached 80% confluence, serum-containing medium was aspirated, the cells were washed gently three times with 10 mL per dish of 37 °C pre-warmed DPBS and incubated in 12 mL of serum-free DMEM for an additional 48 h. Cell-conditioned medium was collected and centrifuged at 2000× *g* at 4 °C for 10 min to remove floating cells. The Sup was subjected to 120,000× *g* centrifugation at 4 °C for 30 min to collect a pellet of apoptotic bodies and micro-vesicles. Finally, Sup was transferred into a 10 mL Backman ultracentrifuge tube and centrifuged at 100,000× *g* at 4 °C for 90 min to collect pelleted exosomes. The exosomes fraction was washed in 10 mL PBS and centrifuged again at 120,000× *g* at 4 °C for 90 min to remove any particles. The EV-depleted Sup was concentrated into desired volume using Amicon^®^ Ultra-4 Centrifugal filters, Ultracel-50K (UFC805024, Millipore Sigma, Burlington, MA, USA) for subsequent analyses.

### 2.5. Trypsin Digestion

Approximately 5 μg of isolated exosomes collected from a conditioned medium of MDA-MB-231 cells were resuspend in a final volume of 0.1 mL of PBS with indicated concentrations of trypsin for 30 min at 37 °C with moderate agitation (Thermomixer 5436, Eppendorf, Hamburg, Germany). In comparison, 30 mL of EV-depleted Sup was concentrated to 0.1 mL using Amicon^®^ Ultra-4 Centrifugal filters, Ultracel-50K (UFC805024, Millipore Sigma, MA, USA), and incubated in PBS with indicated concentrations of trypsin for 30 min at 37 °C with moderate agitation. Purified human rHsp90α protein with indicated concentrations was used in a trypsin digestion assay as a positive control.

### 2.6. Western Immunoblotting Analysis and Band Quantitation

The presence of Hsp90α, β-actin, CD9, GRASP55, GRASP65, Atg5, Atg7, Atg16, Beclin1 and LC3 under various experimental conditions was verified using Western immunoblotting analysis. Total cell lysates were equalized using a BCA protein assay kit (Thermo Scientific). Sample buffer (with 10% β-mercaptoethanol) was added to samples in a 1:3 ratio and boiled for 5 min. The cell lysate, exosomes and EX-depleted Sup fractions were separated by SDS-PAGE and transferred to a nitrocellulose membrane. Ponceau S solution (0.2%) staining was used to confirm the transfer efficiency. The primary antibodies used against the indicated proteins were as described above. Secondary anti-rabbit IgG (AP307P, Millipore Sigma, MA, USA) and anti-mouse IgG (sc-516102, Santa Cruz, TX, USA) were used as instructed by the manufacturers. The intensity of protein bands was quantitated using ImageJ software.

### 2.7. Cell Survival and Growth Assay

Six identical 15-cm cell culture dishes of cells with 50% confluence were prepared and serum-starved for 16 h. Then, all six dishes of cells were washed in PBS 3 times to remove floating cells and changed to fresh serum-free medium. Cells in one culture dish were lifted using trypsin and the viable cell number counted as the starting point on day 0. This step was repeated every 24 h for five days. The average viable cell number in triplicates at each time point (0 h, 24 h, 48 h, 72 h, 96 h, 120 h) were plotted and the cell viability curve established.

### 2.8. Animal Model and Tissue Samples

Hsp90α^−/−^ C57BL/6 mice were generated by CRSPR-cas9 technology, as previously reported [25]. All experimental animal protocols were approved by the USC Institutional Animal Care and Use Committee. Experimental procedures followed the federal guidelines for the care and use of laboratory animals (US Department of Health and Human Services, US Department of Agriculture). Sections of pig skin wounds, control and bleomycin-treated mouse lung, and control and tumor-bearing mouse liver were obtained from our own stocks, Dr. Beiyun Zhou and Dr. Bangyan Stiles, respectively, and immune-stained with a monoclonal antibody against Hsp90α (without cross reactions for Hsp90β).

### 2.9. Wound-Healing Assay

Full-thickness, 8 mm wounds were created using a punch biopsy and surgical scissors, as previously detailed [19,22]. For topical treatments, rHsp90α protein (30 μg per wound) was mixed in a 1:1 ratio with 15% sterile CMC in a final volume of 100 μL (300 μg/mL). Digital photographs were taken individually of the wounds with a metric ruler next to them on the indicated days from a fixed distance using a preset tripod. Biopsies were collected on indicated days and stored in 10% paraffin for sectioning. Photographs of 15 randomly selected images per condition were examined using planimetric measurements for objective evaluation for wound-closure rates. The area of an open wound was calculated as height (cm) × width (cm) to give the cm2 of the wound. Means of more than five wounds were used for the presentation. The area of an open wound on a given day was measured and compared to the area of the wound on day 0 from the same animal, using the software AlphaEase FC, version 4.1.0 (Alpha Innotech Corporation, Miami, FL, USA), as described previously [19,22].

### 2.10. Histology and Immunohistochemistry

The histological and immunohistochemical analyses were carried out for wedged biopsies measuring 1 cm × 1 cm taken on the indicated days for skin wounds, lung, liver and skin biopsies without or with injury or tumor-growth stress. All tissue samples were fixed in 10% formalin (VWR, Randor, PA, USA), and placed in paraffin blocks for sectioning. Immunohistochemical analyses were conducted with anti-Hsp90α antibody. Fifteen randomly selected images of each condition were visualized under a microscope. To carry out semi-quantitation of the staining (blue boxes in Figure 1A), we used Gabriel Landini’s color deconvolution plugin for ImageJ analysis. Using the Image > Color Deconvolution menu, H DAB as the vector and Color 2 as the DAB image, measurements were carried out to convert intensity to optical density (optical density = log (max intensity/mean intensity, where max intensity = 255 for 80-bit images).

### 2.11. Statistical Analysis

Data on animal wound healing were based on at least three independent and corroborating experiments. Data are presented as mean ± standard deviation (SD). Statistical significance was determined by two-tailed Student’s *t* test and two-way analysis of variance (ANOVA) or one-way repeated measures analysis of variance (RMA). Confirmation of a difference as statistically significant requires rejection of the null hypothesis of no difference between means obtained from replicates sets. A *p* value equal to or less than 0.05 was considered statistically significant [19,20,21,22].

## 3. Result

### 3.1. Massive Deposition of eHsp90α by Wounded Tissues and by Cultured Cells under Stress

We proved that secretion of eHsp90α is triggered by cells under stress in vitro and by tissue damage in vivo. First, we tested if eHsp90α is secreted by different types of cells under wound-healing stress signals: TGFα (for keratinocytes), hypoxia (for dermal fibroblasts) and nutrient deprivation (on a randomly selected cell type). It is shown in Figure 1A that Hsp90α was undetectable from CM of untreated cells (panel a–c, lanes 1). In contrast, each of the three types of wound stress dramatically triggered eHsp90α secretion into the CM of the cells (panel a, lane 3, panel b and c, lanes 2). It is pointed out that CMs from equalized cell numbers were loaded on the SDS-PAGE, since there is no universal protein marker for proteins secreted by different cells as a loading control. As previously reported, EGF stimulation triggers eHsp90α secretion less strongly than TGFα in the same cells even though they both bind to the same EGFR (panel a, lane 2 vs. lane 3), consistent with the report that TGFα, but not EGF, is dramatically induced by skin wounding [23]. To confirm the finding in vivo, we used a well-characterized pig wound-healing model. As shown in Figure 1B, a wedge biopsy of a full-thickness excision wound on day 0 showed undetectable staining of a monoclonal anti-Hsp90α antibody (panel d), whereas a biopsy of the same wound on day 4 showed massive staining of the antibody (panel e, as indicated by the red arrows). The staining intensity was quantitated by Gabriel Landini’s color deconvolution plugin and NIH ImageJ (ImageJ) analysis as optical density (OD) (with dashed blue boxes). The location of the epidermis is shown with anti-keratin antibody staining of similar biopsies (Figure 1C, panel f) to better appreciate the location of anti-Hsp90α antibody staining (shown in Figure 1B). It is pointed out that the continuous staining in the dermis of the skin could not be due to fibroblast cells being sectioned at the middle, since skin dermis is mostly composed of collagens with scattered individual cells. To confirm the finding in wounded pig skin, we carried out studies using biopsies from a bleomycin-injured lung, a widely used model for studying lung fibrosis in mice, and biopsies from an HFD (high fat diet)-induced live tumor model in mice. We found that uninjured lung tissue showed little anti-Hsp90α antibody staining. In contrast, the bleomycin-injured lung showed massive anti-Hsp90α antibody staining in the entire section of the biopsy. Similarly, increased anti-Hsp90α antibody staining was detected around the HFD (high fat diet)-induced liver tumors in mice (see page 1, Appendix A). Thus, secretion of eHsp90α occurs in vitro and in vivo under stress.

### 3.2. Majority of eHsp90α Is Not Associated with Secreted Extracellular Vesicles (EV)

Previous studies reported that eHsp90α gets secreted via exosomes and exomeres [16,17,18] but did not provide any data of quantitation for its physiological relevance. Therefore, we decided to investigate what fraction of CM contains the functional concentration of eHsp90α, i.e., 100–300 μg/mL reported to promote maximum wound healing in vivo. Among the four selected cell lines that showed secretion of eHsp90α under serum starvation, as shown in Figure 2A, we chose MDA-MB-231 cells that constitutively secreted eHsp90α (panel a, lane 2) and showed no detectable cell death under serum-free conditions for an initial 72 h as our model, for the reason of convenience. Hela and 293T cells secreted slightly higher and keratinocyte lesser amounts of eHsp90α than MDA-MB-231 cells (lanes 1, 3, 4). The lower secretion of eHsp90α by human keratinocytes was expected, since the primary human skin cells require a combination of multiple stress signals, i.e., serum-free plus hypoxia, to achieve maximum secretion of eHsp90α [23,24]. We then separated CM of MDA-MB-231 cells into microvesicle (MV), exosome (Exo) and supernatant (Sup) fractions following a modified procedure as illustrated in Figure 2B. The three fractions were analyzed by western immunoblotting analysis with an anti-Hsp90α antibody. It is pointed out that the reason for loading differential proportions of the different CM fractions on SDS gel was the realization that the distributions of eHsp90α among the fractions drastically vary. As indicated in Figure 2C, the amount of eHsp90α in MV and Exo fractions even from an entire 15 cm dish showed much less than the amount of eHsp90α in Sup from 1/10 of a 15 cm dish (panel b, lanes 1 and 2 vs. lane 3). The exosome marker CD9 was included in the SDS-PAGE to indicate successful fractionation by sequential centrifugations (panel c, lane 2). Based on data from ImageJ scanning, the distributions of eHsp90α were 2% in the MV fraction, 3% in the Exo fraction and 95% in the (EV-depleted) Sup fraction (Figure 2D). To test whether the majority of eHsp90α in CM is “wrapped” inside lipid vesicles, we found that eHsp90α in CM was as sensitive to trypsin digestion as recombinant Hsp90α protein (Figure 2E, panel d vs. panel e). Therefore, most eHsp90α proteins appeared to be “naked” in CM. To estimate the amount of eHsp90α in CM, we utilized known amounts (μg) of rHsp90α protein as the reference to measure the actual amount of eHsp90α (μg/cells) in the total CM samples from three independent cell cultures. The three CM samples, all from 5 × 10^6^ cells, were immunoblotted together with increasing amounts of rHsp90α protein with an anti-Hsp90α antibody (Figure 2F). The ImageJ data of the rHsp90α protein bands were used to create a linear trendline, as illustrated in Figure 2G. Based on the equation of the linear regression curve, the average amount of eHsp90α protein among the three experiments was 0.85 μg in CM of 5 × 10^6^ MDA-MB-231 cells (under the conditions of serum-free stress for 48 h, where eHsp90α secretion reaches a plateau). Therefore, the distributions of eHsp90α in MV, Exo and Sup are 2%, 3% and 95% of the 0.85 μg from 5 × 10^6^ cells. These numbers will guide the calculation of secreted eHsp90α in tissues as below.

### 3.3. Establishment of Cell-Number-to-Interstitial-Fluid-Volume (CIF) Ratio for Human Tissue In Vivo to Predict Physiological Concentration of a Protein Target in CM of Cells In Vitro

We asked the critical question that has never been asked before “what is the relevance of the 0.85 μg eHsp90α from CM of 5 × 10^6^ cells in vitro to the eHsp90α in human tissues in vivo?” While current investigations of secreted proteins always start with CM of the cells [10,13], the huge discrepancy between the ratio of the medium volume to the cell number in a culture dish and the similar ratio of the interstitial fluid volume to the surrounding cell number in tissues (the physiological environment) has never been taken into account for physiological relevance of a protein target in CM. For instance, the concentration of a secreted protein in a standard 10 cm culture dish (~3 × 10^6^ cells with 10 mL medium) is at least several hundred times more dilute than the concentration of the same protein in interstitial fluid in human tissues. In theory, utilizations of CM, such as functional tests in animal models, should be concentrated according to the physiological ratio. To investigate this previously overlooked problem, as shown Figure 3A, three different types of cells were pooled into 1.5 mL Eppendorf tubes all to reach the cubic volume of 0.1 cm^3^ with 0.1 mL liquid phase. The cell numbers of the three cell lines in this cubic volume were 4.5 × 10^7^, 1.8 × 10^7^ and 2.7 × 10^7^, respectively, with an average ~3 × 10^7^. Therefore, the projected cell number in 1 cm^3^ volume would be ~3 × 10^8^ with 1 mL of medium. However, to culture 3 × 10^8^ cells, thirty 15 cm dishes with a total of 600 mL of medium would be required (assuming 1 × 10^7^ cells in each 15 cm dish with 20 mL of culture medium). Therefore, the ratio of medium volume in the culture to the interstitial fluid volume in tissues with the same number of cells is 600:1. In other word, the concentration of a secreted protein in CM is several hundred times more dilute in comparison to its projected concentration in the 3-D volume. Accordingly, the projected concentration of a secreted protein in a 3-D tissue should be 600-fold higher than its measurable concentration in CM for the same number of cells that secrete this protein.

Therefore, in order to investigate the physiological relevance of the 3-D measurement in vitro, we calculated the ratio of cell number vs. interstitial fluid volume (CIF) for human tissues for the first time. Briefly, for a 70 kg human being, as shown in Figure 3B, the body fluids can be classified into intracellular fluid and extracellular fluid, with a total volume of 42 L. The intracellular fluid resides in the cytoplasm of cells and is estimated at around 28 L. The extracellular fluid has a total volume of 14 L, including the 3 L of plasma (blood) in circulation, the rest being 11 L of interstitial fluid in tissues [26,27,28]. Then, the total body volume and the total number of cells in the body are 70,000 cm^3^ and 3.7 × 10^13^, including 2.5 × 10^13^ blood cells and 1.2 × 10^13^ non-blood tissue cells, respectively [29,30]. Therefore, the human body parameters shown in Figure 3B allowed us to calculate the ratio of the cell number to the surrounding interstitial fluid volume in tissues. As shown in Figure 3C, since 1 cm^3^ tissue is calculated to contain approximately 1.7 × 10^8^ cells and 0.15 mL of interstitial fluid, the human CIF ratio is estimated at 1 × 10^9^ cells/mL of interstitial fluid. Based on 0.85 μg eHsp90α/5 × 10^6^ cells from cell culture (see Figure 2), the amount of secreted eHsp90α in wounded human tissues should be 187 μg/mL, which consists of 178 μg/mL of eHsp90α in EV-depleted Sup (95%) and 9.35 μg/mL of eHsp90α in EVs (5%), as calculated in Figure 3D. Remarkably, the amount of 178 μg/mL of eHsp90α in EV-depleted Sup perfectly matches the previously reported 100–300 μg/mL of rHsp90α protein that promotes maximum wound healing in both mouse and pig wound-healing models [19,20,21,22]. In contrast, secreted exosomes are not sufficient for providing the functional concentration of eHsp90α to promote wound healing. In conclusion, we have established the human CIF ratio in vivo that accurately predicts the tissue concentration of a secreted protein from CM. Using eHsp90α as a marker, we will prove the usefulness of CIF as follows.

### 3.4. Secretory Autophagy, Not Exosomes, Supplies Functional Concentration of eHsp90α for Wound Healing

The 100–300 μg/mL range of concentrations for human recombinant Hsp90α protein (hrHsp90α) to promote maximum wound healing has been accepted as the guideline for therapeutic development (US patents 8,207,118 and 8,455,443). The unanswered question here is what secretion pathway supplies the functional concentrations of eHsp90α to promote wound healing. Since EVs could not quantitively be the source of eHsp90α for wound healing, we asked what EV-independent pathway mediates the secretion of the 95% eHsp90α from CM. The finding that the eHsp90α in CM was “naked” prompted us to investigate other UPS (unconventional protein secretion) pathways. After analyses of the four UPS pathways, as listed in Figure 4A, we concluded that eHsp90α does not meet the requirements for the membrane pore- (type I), ABC transporter- (type II) and Golgi bypass signal peptide- (type IV) mediated pathways for secretion. The most likely mechanism is the type III UPS, the so-called organelle-based secretion pathway, which includes two independent sub-pathways of (i) exosome-mediated and (ii) autophagosome-based secretory autophagy [31,32]. The secretory autophagosome (in contrast to non-secretory/degradative autophagosomes that fuse with lysosomes) is known to fuse with the plasma membrane and release cargos by exocytosis to extracellular environment as naked molecules [33].

There are two parallel upstream regulators of the secretory autophagy pathway [33,34]. First, we tested the family of the Golgi reassembly-stacking protein genes, GRASP-55 and GRASP-65, using GRASP-55- and GRASP-65-knockout (KO) Hela cell lines [35,36]. Figure 4B shows the evidence of GRASP-55 KO (panel a, lane 2) and GRASP-65 KO (panel b, lane 3) in comparison to the unchanged intracellular Hsp90α (panel c). Interestingly, the amount of secreted eHsp90α was selectively reduced in GRASP-55-KO cells by approximately 50% (panel d, lane 2 vs. lane 1, as indicated by a red arrow), and remained unchanged in GRASP-65-KO cells (lane 3). This inhibition of eHsp90α secretion was not due to global inhibition of protein secretion by the cells, since the overall protein secretion by GRASP-55-KO cells was indistinguishable from the parental or GRASP-65-KO cells, as shown by Coomassie blue staining of the total CM of the three cell lines (Figure 4C). To verify that the inhibition of eHsp90α secretion was specifically due to the absence of GRASP-55, we carried out a rescue experiment by re-expressing a GFP-GRASP-55 cDNA into the GRASP-55-KO cells. As shown in Figure 4D, the parental Hela cells express abundant GRASP-55 (panel f, lane 1) and the GRASP-55-KO cells lack any detectable GRASP-55 (lane 2). Lentiviral infection-mediated re-expression of GFP-GRASP-55 shows a similar level to the endogenous GRASP-55 (lane 3 vs. lane 1). Under these conditions, as shown in Figure 4E, GFP-GRASP-55 rescued the lost secretion of eHsp90α in GRASP-55-KO cells (panel i, lane 3 vs. lane 2). The intracellular Hsp90α (panel h) and β-actin (panel j) were included as sample loading controls. To confirm the finding in different cell lines, we carried out lentiviral shRNA-mediated knockdown of GRASP-55 and GRASP-65 in MDA-MB-231 cells. As shown in Figure 4F, downregulation of GRASP-55, but not GRASP-65, partially reduced secretion of eHsp90α (panel l, lane 2 vs. lanes 1 and 3, as indicated by a red arrow). Similarly, as shown in Figure 4G, downregulation of GRASP-55 (panel n, lane 2 vs. lane 1) partially reduced the amount of secreted eHsp90α (panel n, lane 2 vs. lane 1, as indicated by a red arrow) in primary human keratinocytes. Intracellular Hsp90α (panel o) and β-actin (panel p) were included as loading controls. This finding is consistent with the previous reports that GRASP proteins regulate autophagy-mediated secretion of IL-1β [36] and released IL-1β is in the form of a soluble protein rather than being contained within a vesicle [37].

Next, we investigated the autophagosome-regulating (AR) family of genes, including initiation gene Beclin1, elongation genes Atg5 and Atg16 and maturation gene LC3, with Atg7 as a control. As shown in Figure 5A, lentiviral infection-delivered shRNA effectively downregulated the five genes (lanes 1 to 10) in MDA-MB-231 cells. Under these conditions, as shown in Figure 5B, the surface marker of the late autophagosomes, LC-3 (or LC-3B), was dramatically reduced (lanes 2–5 vs. lane 1), indicating disruption of autophagosome formation. GRASP-55 shRNA was included as negative control for specificity (lane 6). Similar to the partial effect of GRASP-55 KO, we found that downregulation of Beclin1, Atg5, Atg16 or LC3A partially blocked the secretion of eHsp90α (Figure 5C, lanes 2, 4, 5 and 6 vs. lane 1), whereas downregulation of Atg7 slightly enhanced secreted eHsp90α (lane 3). Considering both partial inhibitions of eHsp90α secretion, we tested whether double knockdowns of both GRASP55 and AR genes have addictive inhibitions of eHsp90α secretion. As shown in Figure 5D, individual downregulation of GRASP55 and Atg5 each partially reduced secretion of Hsp90α (panel f, lanes 2 and 3 vs. lane 1). However, double knockdowns of both genes almost completely blocked Hsp90α secretion (lane 4). We did not expect 100% blocking of eHsp90α secretion, because the level of gene downregulation was around 80–85%.

To confirm the above findings in human keratinocytes, the essential cell type for wound healing, we repeated the key experiments using human keratinocytes. Similarly, we separated CM of human keratinocytes into MV, Exo and Sup fractions, as illustrated in Figure 2B. The differential loadings of the three fractions on SDS gel were due to drastic variations of eHsp90α among the fractions and made sure to achieve simultaneous visualization of eHsp90α from all three fractions by Western immunoblotting analysis with anti-Hsp90α antibody. As indicated in Figure 6A, the amount of eHsp90α in MV and Exo even from eight times more cells than in Sup showed dramatically less eHsp90α than in Sup (panel a, lanes 1 and 2 vs. lane 3). The exosome marker CD9 was included in the SDS-PAGE to indicate successful fractionation by sequential ultracentrifugation (panel b, lane 2). Based on the data from ImageJ scanning, the distributions of eHsp90α are 0.6% in the MV fraction, 3.4% in the Exo fraction and 96% in the EV-depleted Sup fraction (Figure 6B), which is compatible with MDA-MB-231 cells (see Figure 2D). Similar to the finding in MDA-MB-231 cells, double knockdowns of both GRASP55 and Atg5 expression completely blocked eHsp90α secretion (Figure 6C). Taken together, we concluded these findings as schematically illustrated in Figure 6D. The amount of EV-mediated secretion of eHsp90α, as previously reported, is insufficient for executing the function of promoting wound healing. Instead, the secretory autophagy pathway supplies the 100–300 μg/mL dosage range of secreted eHsp90α for optimal wound healing in vivo.

The CIF ratio-predicted 187 μg/mL of eHsp90α in injured tissues matches perfectly with the finding of previous studies that 100–300 μg/mL of topically applied hrHsp90α maximumly promotes wound healing in vivo [19,20,21,22]. However, no previous study has shown that endogenous eHsp90α is essential for wound healing, since this question could only be answered by Hsp90α-KO mice, which have been recently created in our laboratory [25]. As shown in Figure 7A, unlike Hsp90β-knockout being embryonically lethal in mice [38], Hsp90α-knockout mice are phenotypically normal, supporting the well-accepted understanding that Hsp90β is essential for life and Hsp90α is a stress-responding factor for tissue repair. As shown in Figure 7B, 8 mm × 8 mm full thickness dorsal wounds were created (A, panels a and b) and representative images of wound closure were compared between wild-type and Hsp90α-KO mice (B). The wounds in wild-type mice underwent fast closure, more than 80% on day 10 and 100% on day 12 (panels c, d, e). In comparison, wounds in Hsp90α-KO mice showed a significant delay, less than 70% on day 10 and 80% on day 12 (panels f, g, h). Remarkably, the delay of wound closure in Hsp90α-KO mice was completely corrected by topical application of 300 μg/mL rHsp90α protein (panels j and k vs. panels g and h). The wound-healing images in independent experiments used for statistical analyses are included on pages 7–10 in Appendix A. Quantitation of the wound-closure data is shown in Figure 7C. It is pointed out that, although the main mechanism of mouse skin-wound healing is by wound contraction, completing the final wound closing would require epidermal cell (keratinocyte) migration, called re-epithelialization. Therefore, the finding that the wounds remained open on day 12 in Hsp90α-KO mice indicates that keratinocyte migration was affected. Further support of this notion came from H&E (hematoxylin and eosin) staining of wedged wound biopsies. As shown in Figure 7D, wounds in wild-type mice showed the longest re-epithelialization tongue (Re-epi T, as indicated by the yellow dotted line) (upper panel), wounds in Hsp90α-KO mice showed the shortest Re-epi T (middle panel), and wounds in Hsp90α-KO mice with topically added rHsp90α protein corrected the delay of re-epithelialization (bottom panel). These findings provide genetic evidence that the CIF ratio-calculated amount of eHsp90α is an essential driver of wound healing.

## 4. Discussion

The ~20,000 human protein-coding genes include 16% secreted proteins, 26% membrane bound proteins and the rest intracellular proteins [39]. Biofluids such as plasma and serum are otherwise ideal sources for new biomarker and drug-target discoveries, since they are collections of biological signal molecules from all tissues in the body. Unfortunately, a handful of proteins takes up more than 50% of all the protein contents in human blood, such as albumin, transferrin, and immunoglobulins. Since concentrations of these “background” proteins are more than hundreds of thousands of times higher than any known biomarkers such as growth factors, they shadow any attempts by proteomic studies to achieve clear results. Therefore, CM of cultured cells under serum-free media has been an alternative source to identify new biomarkers and drug targets [40,41]. However, studies have shown that CM studies could suffer from in vitro artifacts of “cell dedifferentiation” and, therefore, any conclusion should be conferred on the corresponding tissue with cautions [42]. Few studies have taken account of the dramatic difference between cell number versus medium volume in vitro and the same number of cells versus surrounding interstitial fluid volume in tissues in vivo in their investigations. For instance, previous utilizations of secreted exosomes in numerous studies, either via topical additions in cultured cells in vitro or IV injection in animals, appeared to be largely random and unexplained in reference to the original volume of the CM, let its alone in vivo relevance.

In this study, we have discovered for the first time the CIF ratio, which is approximately 1.1 × 10^9^ cells versus 1 ml of interstitial fluid for human tissues, as in vivo guidance to evaluate possible physiological relevance of the secreted proteins in a CM with known *K*m (Michaelis constant) concentrations. According to the human CIF ratio, the volume of 20 mL of culture medium for the approximately 1 × 10^7^ cells in a 150 mm tissue culture dish is at least 600 times larger than the projected volume of interstitial fluid surrounding the same number of cells in human tissues in vivo. Therefore, based on the CIF ratio, the contents of a CM should be concentrated several hundred times prior to subjecting it to preclinical and clinical studies. We have proven that the CIF ratio could (1) serve as a guide for concentrating a CM according to the corresponding cell-number-to-interstitial-fluid-volume ratio in tissue before functional studies and (2) predict the physiological concentration of a target protein from a CM based on its reported *K*m. For instance, based on the previous reports of 100–300 μg/mL of rHsp90α protein promoting maximum wound healing in animal models [19,20,21,22], the CIF ratio accurately indicated that the eHsp90α secreted via the secretory autophagy pathway, not via exosomes, is sufficient for promoting wound healing. Therefore, the CIF ratio could provide a valuable piece of information for investigators to decide if the tedious, time-consuming and fallible verification procedures, following initial mass spectrometry analysis, remain warranted. The application of the CIF ratio is limited to individual secreted proteins from a single cell type, which represents the majority of current studies on CM. The CIF-ratio calculation may not apply if the final concentration of a secreted protein is derived from more than one cell type in the tissue microenvironment. In addition, the CIF ratio could vary, since additional fluid or loss of fluid in an injured tissue area could occur under certain pathological conditions, such as edema and dehydration. Any of these factors may have a significant effect on the authenticity of the proteomic profiles.

Currently, it remains largely unclear how autophagosomes avoid fusion with degradative lysosomes. Instead, they deliver secretory cargos to the cell surface and “spill” them out into the extracellular environment. Our results of single and double knockout of GRAPS55 and Atg5 genes and their effects on eHsp90α secretion suggest that there are at least two parallel and independent pathways, i.e., conventional and unconventional, that the secretory autophagy takes. Nishida and colleagues showed that mouse cells lacking Atg5 or Atg7 can still form autophagosomes. In addition to the conventional macroautophagy that depends on autophagosome biogenesis genes such as Atg5, these authors reported that autophagosomes can also be generated in a Rab9-dependent manner by fusion of vesicles derived from the trans-Golgi with late endosomes. They referred to this Rab9-dependent autophagy as alternative autophagy, as illustrated in Figure 6D. Considering GRASP55 is located in the trans-Golgi, it is possible that GRASP55 participates in the alternative secretory autophagy and the AG genes regulate the conventional secretory autophagy. This conventional–alternative autophagy theory perfectly explains the findings of our experiments that GRASP55 and Atg5 each partially regulate eHsp90α secretion and double downregulation of both genes blocks eHsp90α secretion.

Mammals have two Hsp90 genes: Hsp90α and Hsp90β. While the cellular Hsp90β remains steady-state, the cellular level of Hsp90α is drastically affected by environmental stress signals, resulting in a variation from 1.7% to 9% of the total cellular proteins [43]. Hsp90β gene knockout causes death in cultured cells and in mouse embryos, whereas disruption of the Hsp90α gene shows little effect on in vitro or in vivo homeostasis. Therefore, it has been postulated that Hsp90β is the historically reported “critical chaperone” inside cells, whereas the main function of Hsp90α is in its secreted form for tissue repair and wound healing, albeit lacking direct genetic evidence for the latter. We now provide in vivo evidence that Hsp90α protein gets massively secreted in response to tissue injuries. The delay in wound healing in Hsp90α-knockout mice was fully rescued by topically added rHsp90α protein, demonstrating for the first time an essential role for extracellular Hsp90α during wound healing. Most remarkably, application of the CIF ratio allowed us to rule out the physiological relevance of exosome-associated Hsp90α, instead validating the secretory autophagy pathway that supplies the previous established amount of extracellular Hsp90α for achieving *V*max in wound healing in animal models. We speculate that the two main ischemic stress signals in an injured tissue, nutrient paucity and hypoxia, trigger cellular autophagy, leading to the formation of autophagosomes. Autophagosomes are known to have two independent fates, degradative autophagy (autophagosome fusion with lysosomes for degradation) and secretory autophagy (autophagosome fusion with plasma membrane to release their contents into the extracellular environment) [32,33]. Since Hsp90α is disqualified for utilizing the classical ER/Golgi pathway and the three sub-pathways of UPS for its secretion, secretory autophagosome is the most likely mechanism for secretion of more than 90% Hsp90α left in EV-depleted Sup.

## Figures and Tables

**Figure 1 cells-13-01280-f001:**
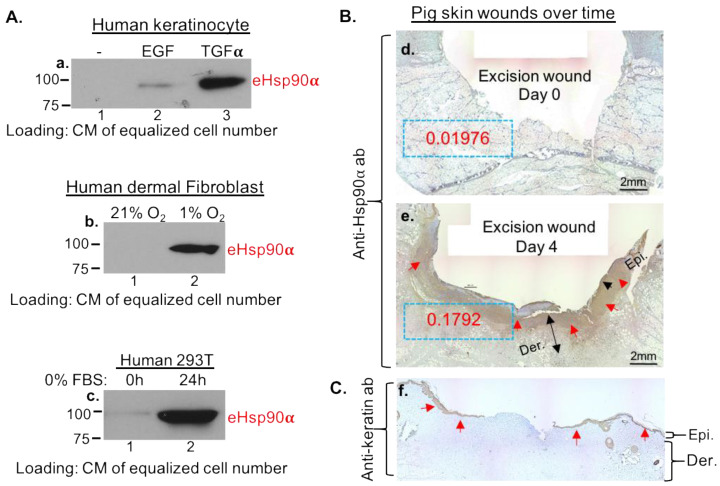
Tissue injury induces massive secretion of eHsp90α in vivo and in vitro. (**A**) Three types of cells, as indicated, were subjected to three kinds of known wound-stress signals: TGFα, hypoxia and nutrient deprivation. CM from the equalized number of each cell type were concentrated and analyzed by western blot analysis with anti-Hsp90α antibodies. (**B**) Wedge biopsies of 1.5 cm full-thickness pig skin wounds on day 0 and day 4 were subjected to anti-Hsp90α antibody staining. The red arrows point out the areas of the specific antibody staining (brown). Gabriel Landini’s color deconvolution plugin and ImageJ analysis were used to quantitate the intensity of the staining as optical density (OD) (dashed blue boxes). (**C**) Similar wedge biopsies were stained with an anti-keratin antibody to indicate the location of epidermis. The data of lung-injury and liver-tumor mouse models, as well as the original films of the western blots, are included on pages 1 and 2 in Appendix A. Epi. Epidermis; Der. Dermis.

**Figure 2 cells-13-01280-f002:**
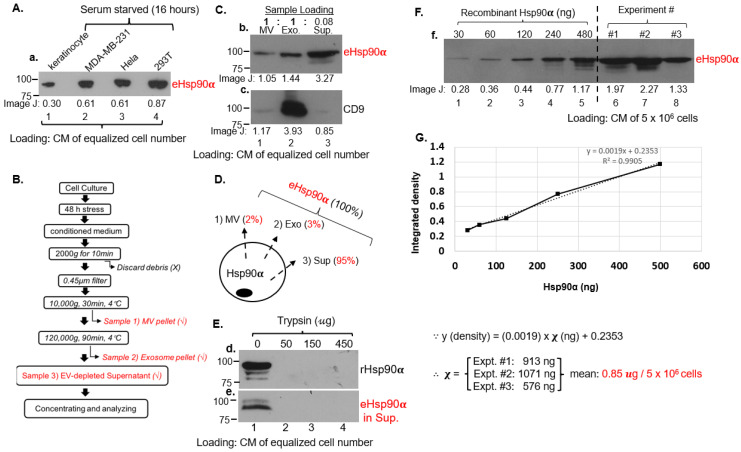
Exosome-independent secretion controls 95% of secreted eHsp90α. (**A**) Comparison of eHsp90α secretion among four cell types under serum-free conditions. CMs of equalized numbers of cells were blotted with an anti-Hsp90α antibody and ECL on films bands quantitated using ImageJ. (**B**) A schematic illustration of a modified CM fractionation protocol, resulting in MV, Exo and Sup fractions for further analyses. (**C**) In order to visualize eHsp90α in all three fractions all at once, the MV and Exo fractions from CM of an entire 15-cm dish and the Sup fraction from 8% CM of a 15-cm dish were loaded on an SDS-PAGE and subjected to western blot analysis with an anti-Hsp90α antibody. The exosomal marker CD9 was included to show the success of the fraction procedures. ImageJ data are shown below the bands. (**D**) Calculation of the percentage of eHsp90α in each of the three fractions. (**E**) EV-depleted Sup was concentrated and subjected to digestion with increasing concentrations of trypsin (panel d) with rHsp90α as the positive control (panel e). Samples were subjected to western blot analysis with an anti-Hsp90α antibody. (**F**) Increasing known amounts of rHsp90α proteins (ng) were loaded together with CMs of five million cells from three independent experiments on an SDS-PAGE and the intensity of anti-Hsp90α body western blotting was compared using ImageJ. (**G**) Based on a chart of band intensity versus ng of rHsp90α protein, an average amount of eHsp90α from three independent experiments was obtained. The original films of the western blots are included on page 3 in Appendix A.

**Figure 3 cells-13-01280-f003:**
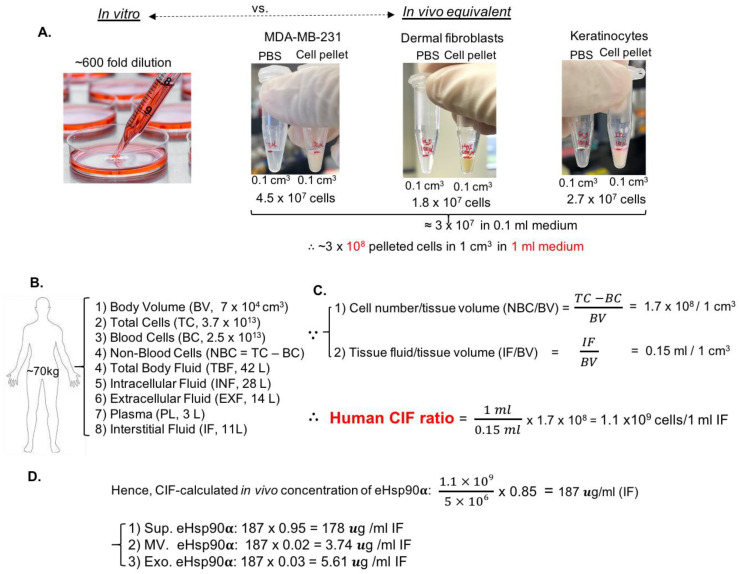
Establishment of human tissue CIF ratio to connect in vitro CM to in vivo interstitial fluid. (**A**) Differences in the medium volume that supports the same number of cells in 2-D versus 3-D cultures. Three different types of cells were collected and pelleted to reach 0.1 cm^3^ and the cell numbers in each tube counted, reaching the final calculation of 3 × 10^8^ cells in 1 cm^3^ volume with a surrounding 1 mL of liquid medium. In comparison, 3 × 10^8^ cells in 2-D culture requires 30 of 15 cm dishes with 600 mL medium to cover the cells. (**B**,**C**) Based on reported human body parameters, the ratio of cell number to volume of interstitial fluid surrounding the cells, or the CIF ratio, was calculated. (**D**) Using the CIF ratio, the projected physiological concentration of secreted Hsp90α in tissue interstitial fluid was calculated, based on the amount of secreted Hsp90α in CM, 0.85 μg/5 × 10^6^ cells (see Figure 2G).

**Figure 4 cells-13-01280-f004:**
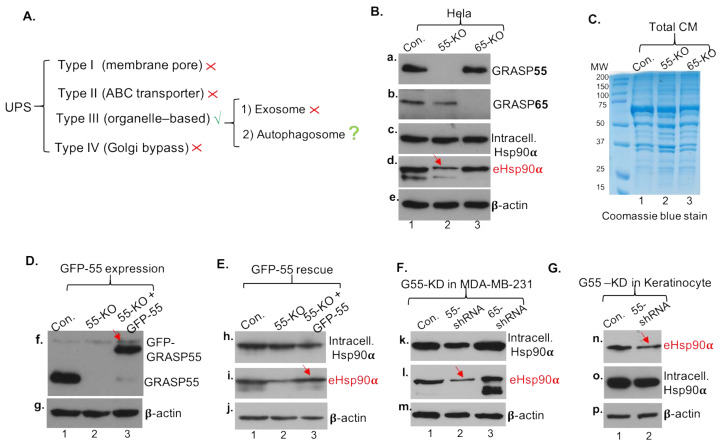
GRASP-55 is necessary and sufficient for partial secretion of eHsp90α. (**A**) Based on known requirements for each of the four types of UPS pathways, the type III pathway is the only possibility for eHsp90α secretion, which does not have any signal peptide and is not found inside vesicles in CM. (**B**) GRASP-55- and GRASP-65-KO Hela cells and their CM of equalized cell numbers were subjected to immune blotting analyses with indicated antibodies. (**C**) The CM from the three cell lines were resolved in SDS gel and stained with Coomassie brilliant blue. (**D**) A GFP-GRASP55 cDNA was expressed in GRASP-55-KO Hela cells to the similar level of the endogenous GRASP-55 in parental Hela cells. (**E**) GFP-GRASP55 rescued the reduced portion of eHsp90α in GRASP-55-KO Hela cells (panel i). (**F**) Lentiviral infection-mediated downregulation of GRASP-55, but not GRASP-65, partially inhibited secretion of eHsp90α in MDA-MB-231 cells (panel l). (**G**) Lentiviral infection-mediated downregulation of GRASP-55 partially inhibited eHsp90α secretion in human keratinocytes (panel n). The original data from the western blot films are included on page 4 in Appendix A.

**Figure 5 cells-13-01280-f005:**
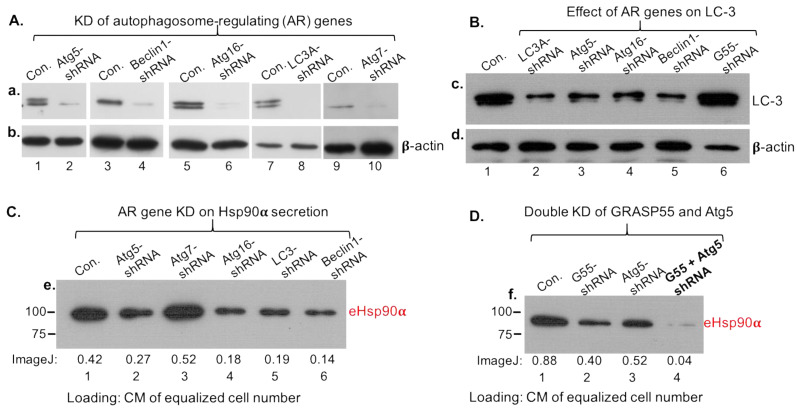
Autophagosome-regulating (AR) genes and GRASP-55 work additively to regulate 95% secretion of eHsp90α. The CIF ratio defines eHsp90α as an essential wound-healing factor. (**A**) Lentiviral infection-mediated downregulation of various AR genes in MDA-MB-231 cells. (**B**) Effect of AR gene downregulation on (late) autophagosome surface marker LC-3. (**C**) The EV-depleted CM or Sup of each of the cell lines were equalized for the same numbers of cells in the culture, concentrated and subjected to western blot analysis with an anti-Hsp90α antibody. (**D**) The EV-depleted CM or Sup of the parental, single or double gene knockout cells were subjected to western blot analysis with an anti-Hsp90α antibody. The original films of the western blots are included on page 5 in Appendix A.

**Figure 6 cells-13-01280-f006:**
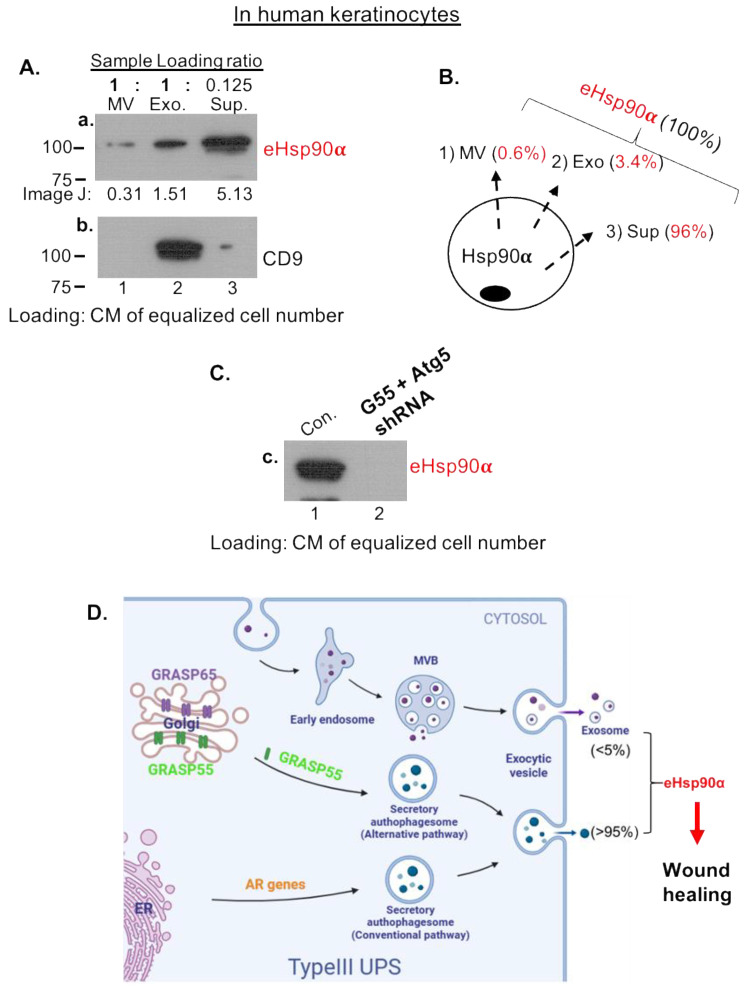
Confirmation of the main findings in human keratinocytes. (**A**) CM of human keratinocytes was separated into MV, Exo and Sup fractions, loaded on an SDS gel with indicated portions, and immunoblotted with anti-Hsp90α antibody. Exosomal marker CD9 was included to show successful fractionation. ImageJ data are shown below the bands. (**B**) Calculation of the percentage of eHsp90α in each of the three fractions based on ImageJ data and portions of loaded fractions. (**C**) EV-depleted CM or Sup from double gene knockout keratinocytes were subjected to western blot analysis with an anti-Hsp90α antibody. (**D**) A schematic summary of the main findings from various cell types. The original films of the western blots are included on page 6 in Appendix A.

**Figure 7 cells-13-01280-f007:**
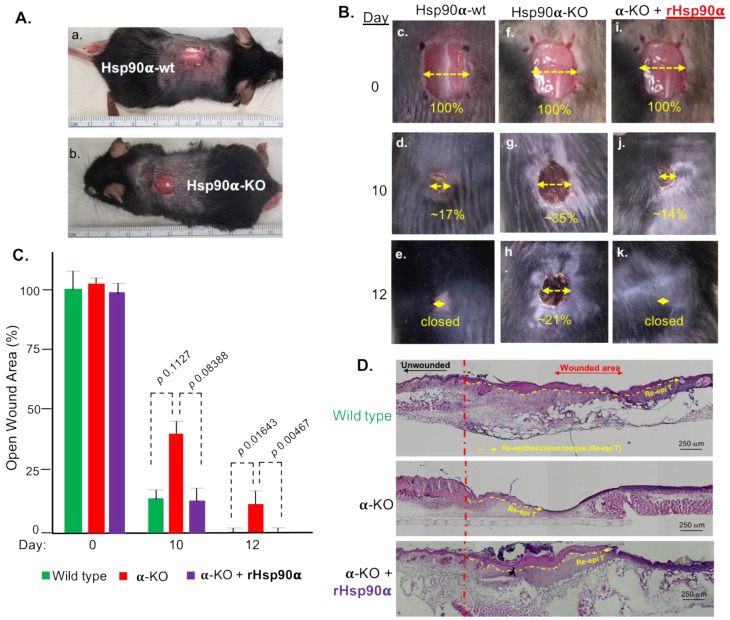
Essential role for eHsp90α from secretory autophagy supply for wound healing in Hsp90α-knockout mice. (**A**) Eight mm (8 × 8 mm) full thickness wounds were created in wild-type and Hsp90α-knockout mice. (**B**) Wounds were treated with or without topical treatment of 300 μg/mL rHsp90α protein (in red). Wound closure was measured as % of the open wound area over time in reference to day 0 wounds (Methods). (**C**) Quantitation of the wound closure as shown in panel B with *p* values, where wounds in Hsp90α-knockout mice still remained open on day 12. (**D**) Section of partial wounds on day 10 were subjected to H&E staining. Red vertical dashed lines divide the unwounded (left) and wounded (right) areas. Yellow horizontal dashed lines show the re-epithelialization tongue (Re-epi T), i.e., epidermal cell migration. Representative images are shown, while additional data of three independent experiments and wound bandages are included on pages 7–10 in Appendix A.

## Data Availability

The data underlying this article are available on request from the corresponding author.

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
