# Peer review of "Discovery of Cell Number-Interstitial Fluid Volume (CIF) Ratio Reveals Secretory Autophagy Pathway to Supply eHsp90α for Wound Healing"

_cells, 2024, doi:10.3390/cells13151280_

Round 1
Reviewer 1 Report
Comments and Suggestions for Authors
The manuscript by Chang et al. encompasses a set of interesting findings concerning the role of secreted Hsp90 alpha in wound healing in vivo, and the Hsp90 alpha KO mice data is quite convincing. The authors further offer intriguing experimental insights into the mechanism of GRASP-55- and Atg machinery-regulated, autophagosome-mediated secretion of the majority of Hsp90 released into the extracellular media upon cellular stress challenge. Despite some data robustness issues, such as the exclusive use of MDA-MB-231 in the experiments of Figure 5 (and also Figure 2, see below), which, as the authors did in Figure 4, should also be reproduced in other cell types, namely in keratinocytes (in TGF alpha-stimulated keratinocytes, is Hsp90 secretion also mediated by autophagosomes?), the autophagosome-dependence for secretion of most of Hsp90, in contrast to previously assumed EV dependence, is an interesting finding that could have significant impact on wound healing research.
In comparison, the CIF ratio story seems secondary to the main findings. Unfortunately, the authors have chosen to make this the focus of the paper. Their extrapolation would fit the paper if presented as a way to justify the actual amount of interstitial Hsp90 required to enhance wound healing in their KO experiments. This is actually another point to be addressed: why only the 300 µg/ml concentration? What happens if the KO mice wounds are treated with lower rHsp90 concentrations (e.g., the 5% (~10-15 µg/mL) present in the MV + EC fraction)? At what concentration does Hsp90 become relevant for wound healing in the KO mice? Considering the dermal tissue context, the extrapolation would also be more relevant if established using the available keratinocyte cell line.
It could then be discussed that similar extrapolations could be performed for other secreted proteins and that, if properly validated, the method could be helpful when analyzing other secretome data obtained using CMs. As the authors briefly acknowledge in their discussion, there are several significant limitations to this sort of extrapolation, including cellular composition, cellular functional heterogeneity, tissue stromal matrix density, abundance and composition, cell/matrix ratios, etc. Thus, focusing the paper’s title, abstract, and discussion on this topic and not on the other interesting findings is somewhat disconcerting. The authors should consider a major revision of the manuscript structure and focus, and clarify the issues raised above, prior to its consideration for publication.
Minor issues:
Culture and handling conditions for 293T cells are absent from the methods section.
Figure 4C: Coomassie bands may reflect mostly residual serum proteins which, as albumin, were left behind and not secreted proteins. Additional controls are needed.
The reference to supplemental information should be at the end of the paper and not in the Figure 5 legend.
Data in Figure 6C requires statistical analysis.
Comments on the Quality of English LanguageGood.
Reviewer 2 Report
Comments and Suggestions for Authors
The study discusses a novel approach to better connect the conditioned medium of cultured cells with the interstitial fluid of tissues for discovering biomarkers and drug targets. The authors introduce the "cell number to interstitial fluid volume" (CIF) ratio. The study uses the secreted protein heat shock protein-90alpha (Hsp90α) as a case study and suggests that the CIF ratio may accurately predicts the concentration of Hsp90α necessary for effective wound healing. The findings show that secretory autophagy, rather than exosomes, is responsible for providing the physiological levels of Hsp90α required for this process. The main assumption is that the CIF ratio enables a more efficient and predictive method for identifying potential biomarkers and therapeutic targets from cell conditioned medium, improving the relevance and accuracy of such discoveries for clinical applications. The study is interesting, but I have few concerns about the results being broadly applicable due to the limited number of cell types used. The following are the key concerns related to this study.
· I have a feeling that different cell types might produce different amounts of proteins. Could it be more appropriate to equalise the different CM versus the total amount of proteins instead of the number of cells?
· Figure 1B. I understand the rationale of the experiment, but the figure is quite unreadable. There is no evident epidermis in the figure of panel d, whereas the same tissue is more evident in panel e. Please explain this difference. In addition, panel d and e appear to be differently stained. This has to be mentioned in the text. What is the precise meaning of the values indicated in the dashed blue boxes? This is not well understood. The dermis in panel e is mislabeled. The double-headed arrow include both the dermis and epidermis (the more basophilic structure).
· The Hsp90 staining is not detectable in figure 1B, panel d. Please improve the either the quality of the magnification of the micrograph.
· The methodology used to calculate the percentage of Hsp90 in Figure 2D is unclear.
· It is unclear why authors keep referring to exosome/exomere as if they are the same structure. This is incorrect. I would not combine these terms.
Comments on the Quality of English LanguageI don't have any particular comment of this part
Round 2
Reviewer 1 Report
Comments and Suggestions for Authors
The authors have made significant improvements to the structure of the manuscript and added key additional data that considerably strengthen their findings and conclusions.
There are only a few minor issues that should be addressed prior to publication:
1. Lines 25-26: “Sequential centrifugations of conditioned medium from relevant cell lines revealed the relative distributions of eHsp90…”
2. Lines 35-36: This sentence must be toned down. Perhaps: “Thus, the establishment of the CIF ratio-based extrapolation method may enable investigators to rapidly predict relevant biomarker targets from conditioned medium data.”
3. Lines 98-100: Perhaps: “We demonstrate that application of the human CIF ratio to secreted Hsp90α from CM of cells predicts a concentration for the secreted factor that perfectly matches…”, since the value of the no-EV fraction was 178 μg/mL and the authors cannot discard the contribution of the EV-contained fraction.
4. Line 103: “for supply over 95% of secreted…”, should be revised to account for both the MDA cells and keratinocyte data.
5. As above, the sentence needs to be a bit toned down: Line 105-106: “Thus, the discovery of the CIF ratio-based extrapolation method may enable investigators to dramatically shorten the screening and identification processes…”
6. Please complete the legend to revised Figure 1 to include the new panel C).
7. Lines 385-386: The statement “We believe that the CIF ratio would significantly shortcut the current procedures to identify biomarkers and therapeutic targets from CM of cells” should be removed from here since it is not a result but an opinion that has been rightfully expressed in the abstract and is later detailed in the discussion.
8. Line 386: Perhaps: “Using eHsp90α as a marker, we further demonstrated the usefulness of the CIF method in biomedical research as follows.”
9. Line 428 and Fig 4C: The reference to serum albumin was not addressed in the text nor in the figure caption, as agreed in the authors’ response letter.
10. Line 626: Please remove “and tumor growth” since it was not addressed in this study.
Author Response
July 21, 2024
Editor-in-Chief
Cells
Dear Editor:
I have enclosed our re-revised (secondary) study entitled “Discovery of [ Cell Number- Interstitial Fluid Volume] (CIF) Ratio Reveals Secretory Autophagy Pathway to Supply eHsp90a for Would Healing” by Cheng Chang, Xin Tang, Axel H. Schönthal, Mei Chen, David T. Woodley, Yanzhuang Wang, Chengyu Liang and myself.
We would like to thank the Reviewer #2 for his/her outstanding scientific insights and work ethic, an excellent example and the high standard for being “a responsible reviewer” (I know that I am not as good myself!). The bottom line is that this reviewer helps make your paper read more professionally and better. We highly appreciate his/her time and efforts.
There are only a few minor issues that should be addressed prior to publication:
- Lines 25-26: “Sequential centrifugations of conditioned medium from relevant cell linesrevealed the relative distributions of eHsp90…”
Done!
- Lines 35-36: This sentence must be toned down. Perhaps: “Thus, the establishmentof the CIF ratio-based extrapolation method may enable investigators to rapidly predict relevant biomarker targets from conditioned medium data.”
Done!
- Lines 98-100: Perhaps: “We demonstrate that application of the human CIF ratio to secreted Hsp90α from CM of cellspredicts a concentration for the secreted factor that perfectly matches…”, since the value of the no-EV fraction was 178 μg/mL and the authors cannot discard the contribution of the EV-contained fraction.
Done!
- Line 103: “for supply over 95%of secreted…”, should be revised to account for both the MDA cells and keratinocyte data.
Done!
- As above, the sentence needs to be a bit toned down: Line 105-106: “Thus, the discovery of the CIF ratio-based extrapolation method mayenable investigators to dramatically shorten the screening and identification processes…”
Done!
- Please complete the legend to revised Figure 1 to include the new panel C).
Done!
- Lines 385-386: The statement “We believe that the CIF ratio would significantly shortcut the current procedures to identify biomarkers and therapeutic targets from CM of cells” should be removed from here since it is not a result but an opinion that has been rightfully expressed in the abstract and is later detailed in the discussion.
Done
- Line 386: Perhaps: “Using eHsp90α as a marker, we further demonstrated the usefulnessof the CIF method in biomedical research as follows.”
Done!
- Line 428 and Fig 4C: The reference to serum albumin was not addressed in the text nor in the figure caption, as agreed in the authors’ response letter.
The journal used the older Figure 4: will point it out to Cells. Text corrected!
- Line 626: Please remove “and tumor growth” since it was not addressed in this study.
Done
Thanks a lot!
Sincerely Yours
Wei Li, Ph.D.
Corresponding Author
Wei Li, Ph.D.
Professor and Director
GMCB (Genetics, Molecular & Cell Biology) and MSS Graduate Programs
USC-Norris Comprehensive Cancer Center and Department of Dermatology
The University of Southern California (USC) Keck School of Medicine
